

# Episodix: a serious game to detect cognitive impairment in senior adults. A psychometric study

Sonia Valladares-Rodriguez[1], Manuel J. Fernández-Iglesias[1,*], Luis Anido-Rifón[1,*], David Facal[2] and Roberto Pérez-Rodríguez[1,*]

[1] Department of Telematics Engineering, Universidad de Vigo, Vigo, Spain
[2] Department of Developmental Psychology, Universidad de Santiago de Compostela, Santiago de Compostela, Spain
[*] These authors contributed equally to this work.

## ABSTRACT

**Introduction**. Assessment of episodic memory is traditionally used to evaluate potential cognitive impairments in senior adults. The present article discusses the capabilities of Episodix, a game to assess the aforementioned cognitive area, as a valid tool to discriminate among mild cognitive impairment (MCI), Alzheimer's disease (AD) and healthy individuals (HC); that is, it studies the game's psychometric validity study to assess cognitive impairment.

**Materials and Methods**. After a preliminary study, a new pilot study, statistically significant for the Galician population, was carried out from a cross-sectional sample of senior adults as target users. A total of 64 individuals (28 HC, 16 MCI, 20 AD) completed the experiment from an initial sample of 74. Participants were administered a collection of classical pen-and-paper tests and interacted with the games developed. A total of six machine learning classification techniques were applied and four relevant performance metrics were computed to assess the classification power of the tool according to participants' cognitive status.

**Results**. According to the classification performance metrics computed, the best classification result is obtained using the Extra Trees Classifier ($F1 = 0.97$ and Cohen's kappa coefficient $= 0.97$). Precision and recall values are also high, above 0.9 for all cognitive groups. Moreover, according to the standard interpretation of Cohen's kappa index, classification is almost perfect (i.e., 0.81–1.00) for the complete dataset for all algorithms.

**Limitations**. Weaknesses (e.g., accessibility, sample size or speed of stimuli) detected during the preliminary study were addressed and solved. Nevertheless, additional research is needed to improve the resolution of the game for the identification of specific cognitive impairments, as well as to achieve a complete validation of the psychometric properties of the digital game.

**Conclusion**. Promising results obtained about psychometric validity of Episodix, represent a relevant step ahead towards the introduction of serious games and machine learning in regular clinical practice for detecting MCI or AD. However, more research is needed to explore the introduction of item response theory in this game and to obtain the required normative data for clinical validity.

Corresponding author
Sonia Valladares-Rodriguez,
soniavr@det.uvigo.es

## INTRODUCTION

The number of people living with dementia around the world is steadily increasing because the primary risk factor is old age (*Winblad et al., 2016*). Their numbers almost double every 20 years, and will reach 131.5 million by 2050 (*Prince, 2015*). Alzheimer's disease (AD) is the most common cause of dementia, as it accounts for an estimated 60% to 80% of cases diagnosed. It is considered a slowly progressive brain disease that begins well before clinical symptoms emerge. Due to this, AD is among the most prevalent medical conditions and one of the most relevant health threats worldwide (*WHO, 2015*).

Diagnosing AD requires a careful and comprehensive health evaluation, including an examination of the patient's mental and functional status, such as, memory, language, visuospatial abilities, executive functions, or living diary activities, among others. Diagnostic procedures for AD are based on clinical procedures, including neuropsychological pen-and-paper tests (*Lezak, 2004*; *Howieson & Lezak, 2010*). These tools are used to perform a cognitive evaluation that may be general or specific to certain cognitive areas (e.g., memory, language, attention, visuospatial capabilities, etc.).

However, classical tests have some relevant limitations, such as being perceived as intrusive (*Chaytor & Schmitter-Edgecombe, 2003*), being influenced by the white-coat effect (*Mario et al., 2009*), providing a late diagnosis (*Holtzman, Morris & Goate, 2011*), lacking ecological validity (*Farias et al., 2003*; *Knight & Titov, 2009*); being strongly dependent on confounding factors (e.g., age, educational level (*Cordell et al., 2013*), practice effect (*Hawkins, Dean & Pearlson, 2004*; *Lezak, 2004*)), or being prone to processing errors due to their manual processing.

Due to the need of alternate ecological mechanisms supporting an early diagnosis of cognitive impairment, the scientific literature discusses several approaches such as the digitalization of classical tests (*Robinson & Brewer, 2016*), the introduction of game-inspired design approaches (e.g., rewards, challenges, simulated environments) (*Tong & Chignell, 2014*) and the implementation of technology-based solutions (e.g., immersive 3D environments, virtual reality, online interactive software applications) (*Parsons et al., 2004*; *Banville et al., 2010*; *Beck et al., 2010*; *Plancher et al., 2012*; *Nolin et al., 2013*; *Nolin et al., 2016*; *Nori et al., 2015*; *Iriarte et al., 2016*), among others. The proposal discussed in this paper relies on the use of gamification techniques, machine learning and the introduction of digital touch devices. More specifically, this paper discusses a psychometric study about a digital game—named Episodix—a video game to assess episodic memory, based on the gamification of the California Verbal Learning Test (CVLT).

The design procedure and a preliminary validation of Episodix was discussed in a previous work (*Valladares-Rodriguez et al., 2017*). Promising results were obtained concerning both Episodix's prediction capabilities and acceptability by target users. Namely,

this tool provided a promising accurate discrimination between healthy participants and individuals suffering mild cognitive impairment, while being highly accepted by target users. Moreover, cognitive areas tackled (*Lezak, 2004*) during game evaluation process are episodic memory, semantic memory and procedural memory. These cognitive areas, and specially the first one, are early markers of cognitive alterations with a relevant diagnostic value for mild cognitive impairment (MCI) and AD (*Juncos-Rabadán et al., 2012*; *Facal, Guàrdia-Olmos & Juncos-Rabadán, 2015*). However, additional research is needed to achieve a complete psychometric validation, and also to improve the predictive ability to detect specific cognitive impairment conditions.

After a thorough revision of the scientific literature in this field (*Helkala et al., 1989*; *Perry, Watson & Hodges, 2000*; *Whitwell et al., 2007*; *Werner et al., 2009*; *Plancher et al., 2010*; *Grambaite et al., 2011*; *Libon et al., 2011*; *Raspelli et al., 2011*; *Aalbers et al., 2013*; *Tarnanas et al., 2013*; *Nolin et al., 2013*; *Fukui et al., 2015*; *Kawahara et al., 2015*) included in our previous methodological review (*Valladares-Rodriguez et al., 2016*), it became apparent that more research was needed to obtain a robust psychometric validation that would eventually support the introduction of cognitive games such as Episodix as a diagnostic tool in everyday clinical practice. In other words, there is a need to empirically evaluate the claim that serious games are a valid means to assess constructs and behaviors with at least the same validity than traditional approaches (*Kato & De Klerk, 2017*).

Thus, additional research was targeted to provide an answer to the following research question: is the Episodix game a valid tool—with a statistical significance for older adults—to discriminate between MCI, AD and healthy individuals? For this, a psychometric validity study of the aforementioned tool was carried out, through a pilot study with a statistically significant sample. On the one hand, the psychometric validity analysis was performed according to the recommendations from the Standards for Educational and Psychological Testing (*AERA, APA & NCME, 2014*). For this, we focused on predictive validity; that is, on how well Episodix predicts cognitive impairment, overcoming the limitations encountered during the preliminary study discussed above (*Valladares-Rodriguez et al., 2017*). Innovative machine learning techniques were applied, which were introduced to support medical prediction and classification related to cancer or heart diseases (*Lehmann et al., 2007*; *Maroco et al., 2011*). The application of this approach is relatively new in cognitive ageing or cognitive impairment studies. A k-fold cross-validation strategy was applied for training and testing the classification models.

The rest of the paper is organized as follows: 'Materials and Methods' introduces the materials and methods utilized in this pilot study, namely test participants' characterization and enrollment, mathematical apparatus used and data analysis performed; 'Results' discusses the outcomes of a pilot experience involving a cross-sectional study participating 74 senior adults; 'Discussion' discusses the psychometric validation process, and finally the main conclusions of this work are summarized.

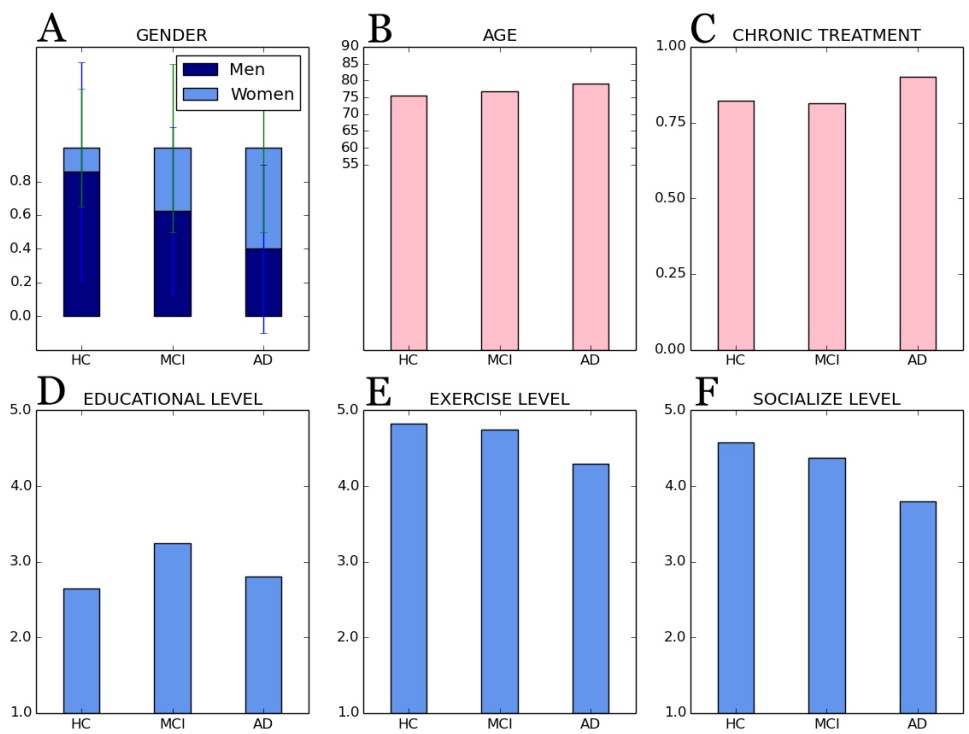

**Figure 1** **General subjects' characteristics according to cognitive group.** Bars are distributed by cognitive groups: (A) gender, men/women; (B) age; (C) chronic treatment, yes/no (i.e., 1/0); (D) educational level, (E) exercise level and (F) socialize level, all of them based on a 5-point Likert scale: 1(never) to 5 (always).

# MATERIALS AND METHODS

## Study design

### Sample description

This study was carried out from a cross-sectional sample of senior adults as target users (cf. Fig. 1). Initially, the sample was composed of 74 people over 55 years old (range = 57–95; average = 77.03; SD = 7.23); from them, 47 women and 27 men (average = 34% males; SD: 0.48), with an average of 6.90 years of education completed. Even though the usual onset age of AD is 65 years old (*Albert et al., 2011*) according to several studies (*Chetelat et al., 2005*; *Rasquin et al., 2005*; *Juncos-Rabadán et al., 2014*), we set the inclusion age in our study at 55 years in order to assess mild cognitive impairment (MCI), an early cognitive damage stage.

All participants were recruited in the Pontevedra province (Galicia, Spain) at the nursing home of Santa Marta in Vigo, at the Association of Relatives of Alzheimer's Patients of Morrazo (AFAMO), and finally, at the Association of Relatives of Alzheimer's Patients and other Dementias of Galicia (AFAGA). The basic inclusion criterion was being 55+ years old, and exclusion criteria included an advanced cognitive impairment, severe motor, hearing or visual disability, and explicit rejection of technology—all issues validated by professionals of

the aforementioned associations. Participants should attend a socio-cognitive workshop in any of the associations indicated above, and no previous educational level or technological skills were required.

During the pilot study, four people were dismissed due to advanced AD; one person passed away; one individual quitted compulsory the socio-cognitive workshop before its completion, and finally four people voluntarily quit the pilot study. Therefore, a cross-sectional sample of 64 subjects eventually completed the study, from them 20 subjects with AD, 16 participants with MCI, and finally, 28 as controls—referred to as the HC group in the rest of this paper.

The study design was approved by the institutional ethics committee (i.e., Galician Ethics Committee for research (Spain), number code 2016/477) and was conducted in accordance with the provisions of the Declaration of Helsinki, as revised in Seoul 2008. Before their involvement in the pilot project, all participants read and understood patient information sheets and gave their written informed consent to use the anonymized data gathered.

Finally, the pilot study was organized in four sessions per subject, lasting 45 minutes each, within a one-month interval. More specifically, three of the sessions were dedicated to game interaction and the last one to classical cognitive evaluation, as discussed below.

### Cognitive assessment

Participants were administered a collection of classical pen-and-paper tests and a questionnaire about socio-demographic aspects such as exercise practice, socialization or present medications to gain some insight on the quality of life of participants and their cognition levels (*Banerjee et al., 2006*).

With respect to the above mentioned tests, MMSE—Mini-metal Examination State (*Cockrell & Folstein, 1987*), the Spanish version of CVLT, California Verbal Learning Test; AD8, Adapted Dementia Screening Interview; ADL, Barthel scale of activities of daily living, and a questionnaire about memory complaints were administered (cf. Table 1). Data gathered using these tests were used as golden standard data and also to initially discriminate subjects suffering MCI from healthy controls. According to standard diagnostic criteria (*Petersen, 2004*; *Albert et al., 2011*; *Juncos-Rabadán et al., 2012*; *Campos-Magdaleno et al., 2017*), MCI would likely be present for participants that would present normal cognitive functions and their capabilities to carry out everyday activities are not compromised, they would refer memory complaints, a low score in the Spanish version of CVLT according to subject's age and educational background, and a level equal or greater than 2 in the AD8 test. Participants who suffered AD —facilitated by participating organizations with a prior medical diagnosis—were not asked to take the Spanish version of CVLT to avoid frustration.

Finally, demographic and lifestyle information collected (cf. Fig. 1) show that social interaction and physical activities decrease with increasing cognitive impairment. Also,

**Table 1** Comparative of participants' cognitive assessment by cognitive group.

| TESTS SCORES | HC | MCI | AD |
|---|---|---|---|
| MMSE | 26.93 ± 2.11 | 25.37 ± 2.55 | 23.00 ± 4.38 |
| RI_AT: Total Trial 1-5 [Spanish version of CVLT] | 47.25 ± 5.83 | 34.62 ± 9.50 | |
| RL_CP: Short Delay Free Recall [Spanish version of CVLT] | 9.92 ± 2.55 | 6.31 ± 1.74 | |
| RL_LP: Long Delay Free Recall [Spanish version of CVLT] | 10.93 ± 2.39 | 5.87 ± 3.07 | |
| AD8 | 1.39 ± 1.47 | 1.75 ± 1.18 | 5.40 ± 1.31 |
| ADL Barthel scale | 102.07 ± 6.39 | 100.00 ± 7.28 | 91.85 ± 11.00 |
| Subjective memory complaints [by participants] | 1.99 ± 0.51 | 1.99 ± 0.57 | 2.47 ± 0.51 |
| Subjective memory complaints [by relatives] | 1.82 ± 0.73 | 1.78 ± 0.41 | 2.46 ± 0.64 |

Notes.

MMSE cut-off score: 9–12 means AD is likely present. Spanish version of CVLT cut-off scores computed according to the results of Total Trial 1–5, Short Delay Free Recall, Long Delay Free Recall, age and years of education (*Albert et al., 2011*; *Juncos-Rabadán et al., 2012*; *Campos-Magdaleno et al., 2017*). Memory complaints (by participants and informants) based on a 5-point Likert scale: 1(never forget) to 5 (always forget). AD8, 0–1: normal cognition; 2 or greater: cognitive impairment is likely to be present. Finally, ADL: 0–20: total dependency; 21–60: severe dependency; 61–90: moderate dependency; 91–99: small dependency; and 100: autonomy.

participating individuals are in most cases subject to chronic medication due to age-related conditions such as hypertension, hypercholesterolemia, or diabetes, and medication increases with age and increasing cognitive impairment.

### Serious game assessment

Gaming sessions consisted on three consecutive phases. Firstly, participants would play the first part of Episodix (cf. A and B in Fig. 2), which target immediate and short-term memory. Then, they would play two additional games to assess semantic and procedural memory (cf. C and D in Fig. 2 and Appendix C with supplementary information). Finally, participants would play the second part of Episodix to assess long-term memory and yes/no recognition capabilities. Note that this approach emulates the administration of CVLT, only using serious games instead of pen-and-paper tests, and breaks from being used to evaluate other cognitive markers.

From a technological point of view, games were developed in Unity (*Unity, 2016*) and were run in a Samsung Galaxy Note Pro (SM-P900) tablet device. Thus, senior adults interacted with a touch version of the tool to enhance its usability. Episodix also offers multi-language support, and all stimuli are additionally presented in textual and audio formats to enhance accessibility.

Finally, serious game assessment was carried out at the same time and location as the compulsory socio-cognitive workshops to guarantee the ecological validity of neuropsychological evaluation by integrating the latter in participating participants' daily routines. This was particularly relevant for participants with cognitive impairment, for whom there is a high prevalence of the white-coat effect (*Mario et al., 2009*).

## Data analysis
### Statistical significance

The sample of senior adults recruited fulfills the statistical representativeness criteria for the senior population in Galicia, Spain. The sample size was computed according to the

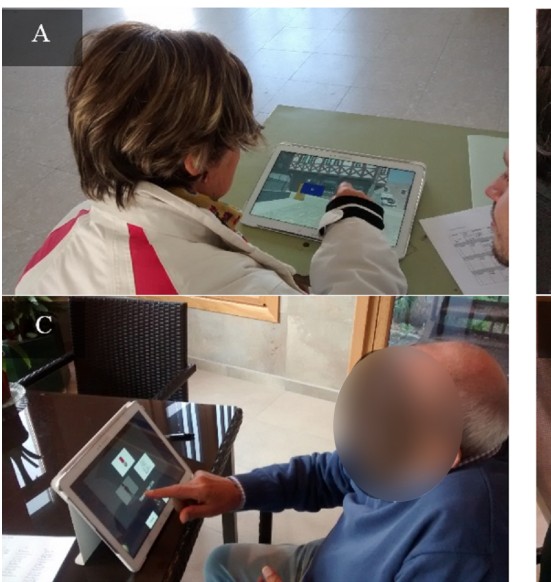
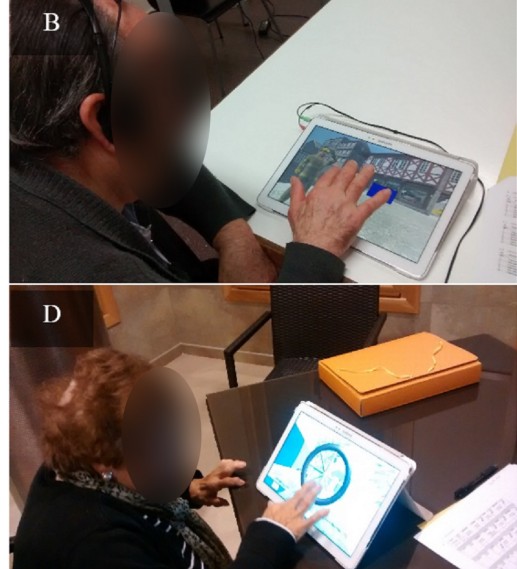

**Figure 2** **Some captures of cognitive game sessions.** (A and B) Episodix, (C) semantic memory game and (D) procedural memory game. Photo permission was required and granted in the patient informant sheet. Photo credit: Sonia Valladares-Rodriguez.

minimum sample for a prevalence study based on proportional stratified sampling (*Barlett, Kotrlik & Higgins, 2001*):

$$n = \frac{N * Z\alpha^2 * p * q}{\varepsilon^2 * (N-1) + Z\alpha^2 * p * q}$$

where $N$ represents the size of the Galician population over 55 years according to the National Statistics Institute; $Z$ is the statistical confidence level (i.e., $1.96 \approx 95\%$); $p$ represents the probability of AD prevalence in Galicia according to the National Statistics Institute; $q$ is the probability of AD's absence (i.e., $q = 1 - p$); indicates admissible error (i.e., 7.5%), which is within the usual values in research studies (i.e., 5%–10%). As a result, a sample of $n = 62$ individuals would be required for the results obtained during the pilot study to be statistically significant.

### Psychometric study

To assess whether Episodix is a valid cognitive game to discriminate between individuals suffering MCI, individuals suffering AD or participants not suffering cognitive impairment, we focused on psychometric criterion validity (*Souza, Alexandre & Guirardello, 2017*), in particular predictive ability. In other words, we tested the validity of serious games by comparing them with golden standard criteria (i.e., classical tests described in 'Cognitive assessment').

In order to do so, recent machine learning classification techniques were applied. Particularly, we selected popular techniques in medical research based on supervised learning (*Lehmann et al., 2007*; *Tripoliti et al., 2010*; *Maroco et al., 2011*), in order to obtain a prediction or classification score about cognitive impairment (i.e., MCI, AD or absence of
cognitive problems). Techniques introduced in the preliminary study were also applied now (i.e., LR, SVM and RF) (*Valladares-Rodriguez et al., 2017*), and three new techniques were introduced (i.e., ET, AB and GB) due to their high efficacy when applied to classification problems.

- LR—Logistic regression. It is a statistical method for analyzing a dataset in which there are one or more independent variables that determine an outcome. The outcome is measured with a dichotomous variable (in which there are only two possible outcomes). This is a simple technique widely used in predictive analysis in the medical field.
- SVM—Support Vector Machines (*Hearst et al., 1998*). This technique is widely used for clustering and classification. It aims at building a model able to predict to which category a new element added to the model would belong. SVM represents each element as a point in an n-dimensional space. Then, different element classes correspond to clusters in that space. Separating hyperplanes, named "support vectors", are computed to discriminate among clusters.
- RF—Random Forests (*Breiman, 2001*). This algorithm, also known as random decision forests, is a perturb-and-combine technique specifically designed for decision trees where random alterations are introduced in the learning process. RF is an ensemble learning method that provides better prediction capabilities tan other decision tree-based methods. In addition, it corrects the latter's' trend to overfitting.
- ET—Extra Trees Classifier (*Geurts, Ernst & Wehenkel, 2006*) is based on unpruned top-down decision trees and are also based on the perturb-and-combine strategy. They are typically used to enhance precision in predictions provided by traditional decision trees. "Extra" stands for extremely randomized trees.
- GB—Gradient Boosting Classifier (*Payan & Montana, 2015*). A technique, based on decision tree learning (https://en.wikipedia.org/wiki/Decision_tree_learning), which produces a prediction model in the form of an ensemble method (https://en.wikipedia.org/wiki/Ensemble_learning) weak prediction models. Namely, it optimizes a cost function over function space by iteratively choosing a function (weak hypothesis) that points in the negative gradient direction.
- AB—Ada Boost Classifier (*Freund & Schapire, 1997*). This machine learning meta algorithm (https://en.wikipedia.org/wiki/Machine_learning; https://en.wikipedia.org/wiki/Meta-algorithm) is adaptive in the sense that subsequent weak learners are tweaked in favor of those instances misclassified by previous classifiers. Moreover, this algorithm is sensitive to noisy data and outliers (https://en.wikipedia.org/wiki/Outlier). For each of the previous algorithms, quality metrics widely used in medical studies were computed:
- F1-score: weighted harmonic average of precision and recall.

$$F1(F1score) = 2 * \frac{P*R}{P+R}$$

- Cohen's kappa coefficient (*Cohen, 1960*) provides a measure of the validity of the classification when confronted with a random result.
- value ranges are interpreted according to the existing consensus (*Landis & Koch, 1977*): 0–0.20 Insignificant (i.e., ML-based classification would not be distinguishable from

random classification); 0.21–0.40 Median; 0.41–0.60 Moderate; 0.61–0.80 Substantial; and 0.81–1.00 Almost perfect;

$$k\,(\text{Cohen's kappa}) = 1 - \frac{(1 - Po)}{(1 - Pe)}$$

where $P_o$ is the relative observed agreement (i.e., accuracy), and $P_e$ the estimated or hypothetical probability of random agreement.

- Recall or sensitivity: the game's ability to identify actually impaired people to avoid false negatives.

$$Recall = \frac{TP}{(TP + FN)}$$

- Precision or ratio of relevant information over total information obtained during the prediction process.

$$Precison = \frac{TP}{(TP + FP)}.$$

About the last two metrics, we computed the precision-recall curve as an indication of classification quality. Note that a larger area below the curve indicates better precision (i.e., game's ability to obtain positive predicted values) and better recall (i.e., game's ability to avoid false negatives). This curve is commonly used in the analysis of information retrieval systems as an alternative to the ROC curve because it provides a more informative picture of the classifier's performance (*Davis & Goadrich, 2006*).

The aforementioned metrics were reported as macro values, because they are more convenient to show the performance of the classifiers with data sets that are not evenly distributed over the available categories. The calculation of macro metrics is based simply on computing the metrics for each label or class and then obtaining its un-weighted average.

Finally, to prevent over-fitting and artificial classification metrics due to the use of the same data for training and testing of classifiers, a k-fold cross-validation strategy was adopted to train and evaluate the four proposed classifiers. Furthermore, the machine learning library Scikit-Learn (*Pedregosa et al., 2011*) was used for all data analytics, under a Python ecosystem. Missing data due to the different difficulty levels achieved by different participants, was addressed by means of substitution by cognitive group (i.e., AD, MCI or HC). This frequently used method replaces the missing data for a given attribute by the mean of all known values of that attribute, for the different cognitive groups (*Batista & Monard, 2003*; *García-Laencina, Sancho-Gómez & Figueiras-Vidal, 2010*).

## RESULTS

This section presents the results of the psychometric study of Episodix. Both aspects were analyzed for the complete sample ($n = 64$) and for the cognitive groups defined HC ($n = 28$), MCI ($n = 16$) and AD ($n = 20$).

### General and cognitive characteristics of participants

Participants (cf. Fig. 2) were characterized according to the parameters below (characteristics are expressed as average values and their standard deviation).

Regarding the general demographic details (cf. Fig. 1) for each cognitive group:

- Age: participants have an average of 75.57 ± 7.14 years old for controls; 76.87 ± 9.33 years old for participants with MCI; and, finally, 79.15 ± 4.91 years old for people with AD;
- Gender: the average of females in the sample was 0.14 ±0.35 for HC group; 0.37 ± 0.50 for females with MCI; and 0.60 ± 0.50 for AD group.
- Chronic treatment (i.e., a dichotomous variable "yes" or "no"): the 0.82 ± 0.39 for healthy participants; the 0.81 ± 0.40 for people with MCI; and a mean of 0.90 ±0.31 AD patients under treatment.
- Educational level (i.e., years of education): 1.34 ± 0.82 for people without cognitive impairment; 1.78 ± 0.41 for MCI participants' group; and 1.80 ± 1.19 mean years of education for people with AD.
- Exercise level (i.e., 5-point Likert scale: 1(nothing) to 5 (a lot)): 3.82 ± 1.09 for controls; 3.75 ± 0.86 for MCI participants' group; and 3.30 ± 0.92 for AD group.
- Socialize level (i.e., 5-point Likert scale: 1(nothing) to 5 (a lot)): 3.58 ± 1.10 for HC participants; 3.37 ± 0.88 for participants with MCI; and 2.80 ± 0.95 for people affected by AD.

With respect to the outcomes of classical testing (cf. Table 1), once again according to each cognitive group:

- MMSE: 26.93 ± 2.11 for HC participants; 25.37 ± 2.55 for people with MCI; and 23.00 ± 4.38 for people affected by AD, over a total score of 30.
- Spanish version of CVLT, according to standard diagnosis criteria. Total Trial 1-5: 47.25 ± 5.83 [HC group] and 34.62 ± 9.50 [MCI group]; Short Delay Free Recall: 9.93 ± 2.55 [HC group] and 6.31 ± 1.74 [MCI group]; and finally, Long Delay Free Recall: 10.93 ± 2.39 [HC group] and 5.87 ± 3.07 [MCI group].
- AD8: in this case the average score for HC was 1.39 ± 1.47; 1.75 ± 1.18 for participants with MCI; 5.40 ± 1.31 for AD participants.
- ADL Barthel scale: 102.07 ± 6.39 for HC individuals; 100 ± 7.28 for MCI participants; and finally, 91.85 ±11.03 for participants affected by AD.
- Subject memory complaints raised by participants about themselves: 1.99 ± 0.51 for HC; 1.99 ± 0.57 for participants with MCI; and 2.47 ± 0.50 for AD participants.
- Subject memory complaints raised by an informant: 1.82 ±0.73 for HC participants; 1.78 ± 0.41 for MCI people; 2.46 ± 0.64 for people affected by AD.

## Psychometric study outcomes

All participants played two complete game sessions, with each session consisting of two Episodix phases separated by semantic & procedural memory games during the break between the two Episodix phases. Obtained datasets were processed with the machine learning algorithms enumerated above, that is, Extra Trees classifier (ET); Gradient Boosting Classifier (GB); Ada Boost Classifier (AB); Support Vector Machines (SVM); Logistic Regression (LR); and Random Forest (RF).

**Table 2** Metrics about psychometric predicted validity of the Episodix game.

| EPISODIX DATA SET | | | Recall | | | Precision | | |
|---|---|---|---|---|---|---|---|---|
| ML algorithms | F1 | k ↓ | HC | MCI | AD | HC | MCI | AD |
| GB: Gradient Boosting | 0,92 | 0,89 | 0,96 | 0,92 | 0,91 | 0,94 | 0,87 | 0,96 |
| ET: Extra Trees | 0,91 | 0,88 | 0,91 | 0,94 | 0,95 | 0,97 | 0,76 | 0,96 |
| RF: Random Forest | 0,89 | 0,86 | 0,86 | 1,00 | 0,95 | 0,99 | 0,66 | 0,96 |
| AB: Ada Boost | 0,81 | 0,76 | 0,79 | 0,76 | 0,96 | 0,91 | 0,58 | 0,92 |
| SVM: Support Vector Machine | 0,80 | 0,75 | 0,80 | 0,75 | 0,93 | 0,91 | 0,55 | 0,92 |
| LR: Logistic Regression | 0,76 | 0,70 | 0,74 | 0,73 | 0,94 | 0,92 | 0,42 | 0,91 |

Notes.

Episodix dataset (191 triples). ML algorithms: ET: Extra Trees classifier; GB: Gradient Boosting classifier; AB: Ada Boost classifier; SVM: Support Vector Machines; LR: Logistic regression; and RF: Random forest. Metrics: F1 score; Cohen's Kappa (i.e., used as index to order best classification); recall and precision, the last two distributed by cognitive group. Experiments were performances with cv-fold cross validation ($cv = 55$) and default configuration in ML algorithms.

Firstly, ML algorithms were applied to the dataset composed of Episodix data, focusing on psychometric predicted validity. Thus, a total of 191 triplets were included in the dataset. The metrics below were computed to estimate the classification abilities (cf. Table 2, sorted according to decreasing k↓):

- F1 score: average values obtained were 0.92 for GB; 0.91 for ET; 0.89 for ET; 0.81 for AB; 0.80 for SVM; and 0.76 for LR.
- Cohen's kappa: the average value obtained for the concordance of the observed participants' classification for each algorithm was: 0.89 for GB; 0.91 or ET; 0.86 for RF; 0.76 for AB; 0.75 for SVM; and 0.70 for LR.
- Recall was computed for each cognitive group:
  - ET: 0.96 for HC, 0.92 for MCI, and 0.91 for participants with AD.
  - GB: in the range of 0.91–0.95.
  - AB: in the range of 0.76–0.96.
  - SVM: in the range of 0.75–0.93.
  - LR: 0.74 for HC, 0.73 for MCI, and 0.94 for participants with AD.
  - RF: in the range of 0.86–1.
- Precision, again by cognitive group:
  - HC: in the range of 0.91–0.99.
  - MCI: in the range of 0.42–0.87.
  - AD: in the range of 0.91–0.96.

The same ML algorithms were applied to a dataset including Episodix, semantic memory and procedural memory data (458 triples). The metrics below were computed to estimate the classification abilities (cf. Table 3, again sorted according to decreasing k↓):

- F1 score: average values obtained were .97 for ET; 0.97 for RF; 0.96 for GB; 0.91 for SVM; 0.84 for LR; and 0.84 for AB.

**Table 3** Metrics about psychometric predicted validity of the Episodix, semantic memory and procedural memory games.

| EPISODIX+S+PDATASET | | | Recall | | | Precision | | |
|---|---|---|---|---|---|---|---|---|
| ML classifier | F1 | k ↓ | HC | MCI | AD | HC | MCI | AD |
| ET: Extra Trees | 0,97 | 0,97 | 0,96 | 0,99 | 0,99 | 0,99 | 0,91 | 1,00 |
| RF: Random Forest | 0,97 | 0,96 | 0,95 | 0,98 | 0,99 | 0,98 | 0,91 | 1,00 |
| GB: Gradient Boosting | 0,96 | 0,95 | 0,95 | 0,94 | 0,98 | 0,96 | 0,93 | 0,98 |
| SVM: Support Vector Machine | 0,91 | 0,90 | 0,92 | 0,82 | 0,99 | 0,90 | 0,87 | 0,98 |
| LR: Logistic Regression | 0,84 | 0,81 | 0,77 | 0,80 | 0,98 | 0,90 | 0,67 | 0,95 |
| AB: Ada Boost | 0,84 | 0,80 | 0,77 | 0,78 | 0,99 | 0,90 | 0,70 | 0,93 |

**Notes.**
Episodix, semantic memory and procedural memory dataset (458 triples). ML algorithms: GB, Gradient Boosting classifier; ET, Extra Trees classifier; SVM, Support Vector Machines; LR, Logistic regression; AB, Ada Boost classifier; and RF, Random forest. Metrics: F1 score; Cohen's Kappa (i.e., used as index to order best classification); recall and precision, the last two distributed by cognitive group. Experiments were performances with cv-fold cross validation ($cv = 55$) and default configuration in ML algorithms.

- Cohen's kappa: the average value obtained for the concordance of the observed participants' classification for each algorithm was: 0.97 for ET; 0.967 for RF; 0.95 for GB; 0.90 for SVM; 0.81 for LR; and 0.80 for AB.
- Recall by cognitive group:
  - HC: in the range of 0.77–0.96.
  - MCI: in the range of 0.78–0.99.
  - AD: in the range of 0.98–0.99.

- Precision, by cognitive group:
  - HC: in the range of 0.90–0.99.
  - MCI: in the range of 0.67–0.93.
  - AD: in the range of 0.93–1.00.

As higher values were obtained for the metrics computed in this case, the precision-recall curve was also computed for the complete dataset including Episodix, semantic memory and procedural memory data. The curve depicts the average of precision and recall, together with the area below the curve, for each cognitive group. In this case, as Fig. 3 shows, a high area under the curve is obtained for all ML algorithms, specially, for GB, RF and ET with an average precision higher to 0.93. All algorithms show a high precision-recall area for the AD group (i.e., value higher to 0.95), followed by the HC group (i.e., value higher to 0.95 except for AB and LR), and finally the MCI group (i.e., value higher to 0.74, except again for AB and LR).

Furthermore, we computed the collection of the 10 most informative variables for classification for both datasets. For this two models of features selection, based on chi-squared and ANOVA $F$-value statistics with similar results. As a consequence, the set of most informative features are (cf. Appendix A for additional details):

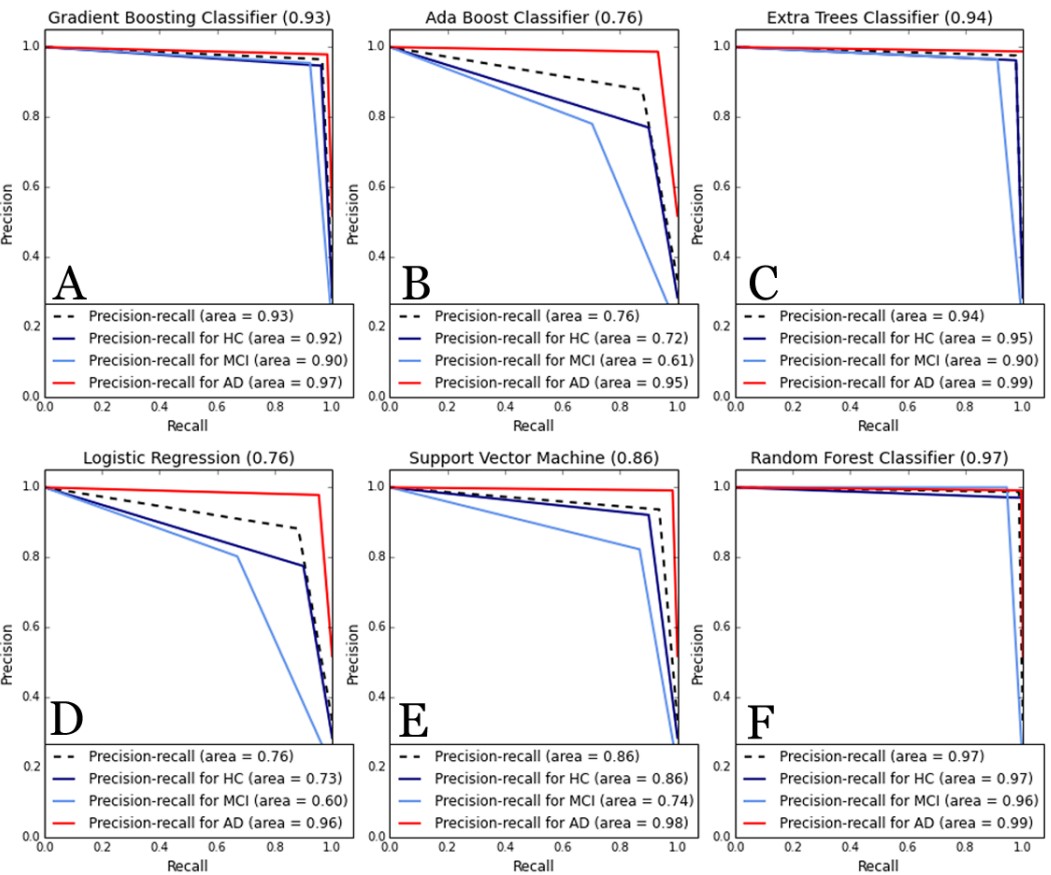

**Figure 3 Comparative of Precision-Recall curve metric to evaluate classifier output.** ML algorithms: (A) GB, Gradient Boosting Classifier; (B) AB, Ada Boost Classifier; (C) ET, Extra Trees classifier; (D) LR, Logistic regression; (E) SVM, Support Vector Machines; and (F) RF, Random forest. Metrics: precision-recall curve average for each cognitive group. Experiments were performances with cv-fold cross validation ($cv = 55$).

- EPISODIX dataset: number of failures during immediate recall phase, short-term recall and long-term recall; number of omissions during long-term recall with semantic clues; and number of repetitions during yes/no recognition phase.
- EPISODIX+S+P dataset: in general, the most informative variables are the same as in the previous case. However, in this case, time duration of procedural memory tasks is included among the 10 most informative variables.

Finally, we applied again the six ML algorithms with a subset including the aforementioned 10 most informative variables for classification for both datasets (cf. Appendix B for additional details). On the one hand, using only the Episodix data set, quality metrics dropped in general terms. For instance, the F1-score dropped to 0.59–0.87 from an initial range of 0.76–0.93. In the same line, average precision-recall area was reduced (i.e., from 0.89–0.72 to 0.54–0.8). On the other hand, if we consider the metrics obtained for the three games using the most informative features, they also dropped slightly. By focusing

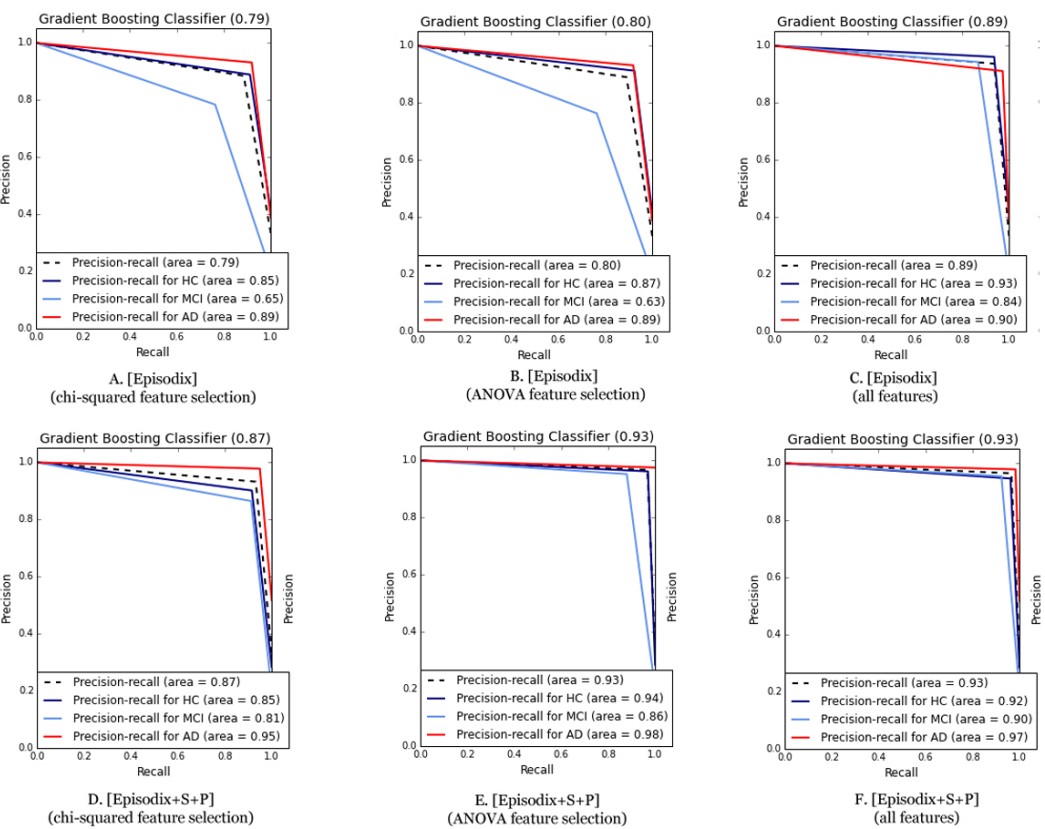

**Figure 4** **Comparative of Precision-Recall curve metric using the best performance algorithm (i.e., Gradient Boosting classifier).** (A) Episodix data set, GB, Gradient Boosting Classifier and chi-squared feature selection; (B) Episodix data set, GB, Gradient Boosting Classifier and ANOVA $F$-test feature selection; (C) Episodix data set, GB, Gradient Boosting Classifier and all features; (D) Episodix+S+P data set, GB, Gradient Boosting Classifier and chi-squared feature selection; (E) Episodix+S+P data set, GB, Gradient Boosting Classifier and ANOVA $F$-test feature selection; and (F) Episodix+S+P data set, GB, Gradient Boosting Classifier and all features.

on the best performance obtained using the Gradient Boosting classifier, F1 changes from 0.96 to 0.93; Cohen's kappa raises from 0.94 to 0.95; and finally, precision-recall area remains the same (i.e., 0.93 for ANOVA $F$-value feature selection). Figure 4 below depicts the comparative of Precision-Recall curve metrics using the Gradient Boosting classifier. The area is larger using all the features (i.e., 81 features for Episodix and 100 features for Episodix+S+P) in comparison to the 10 most informative variables, including subject's age and genre. Furthermore, in the case of ANOVA $F$-value selection, general results are equal, showing a small area in the prediction of the MCI group (i.e., it varies from 0.90 to 0.86).

## DISCUSSION

This article discusses the psychometric study of Episodix, a game-based battery to evaluate cognitive impairment. Previous research stated the preliminary prediction ability of this instrument (*Valladares-Rodriguez et al., 2017*). The justification of this new research is to

overcome the limitations detected in that previous study; in particular, those related to the representativeness of the population sample. Thus, in this case the objective would be to validate from a sound psychometric perspective, the diagnostic abilities of the game supported by supervised machine learning algorithms.

In this new study, a total of 64 individuals completed the pilot experiment from an initial sample of 74. Twenty-eight participants were included in the healthy control group, while 16 people presented MCI and 20 suffered AD. The sample was statistically representative of the size of Galician senior population, which in turn shares demographic characteristics with the population in many other rural areas in Europe. The sample is also representative from a research perspective (*Barlett, Kotrlik & Higgins, 2001*). In particular, according to recommendations for this type of studies (*Hajian-Tilaki, 2014*), the sample utilized satisfies the required total sample size (i.e., 41 < number of participants < 78) for estimating a specificity between 0.90–0.95, for a marginal error of 0.07 and prevalence of 0.1.

On the other side, participating senior adults felt comfortable during the evaluation sessions embedded in their daily routines, which makes this tool an innovative ecological mechanism supporting an early diagnosis of cognitive impairment. Also, no technological impediments were detected insofar usability is concerned due to the introduction of touch devices. To sum up, this novel approach to neuropsychological evaluation overcomes the limitations of classical tests, as it avoids the white-coat effect, confounding factors or errors common while computing final scores. All of these issues are even more relevant when target users are senior adults.

After a preliminary psychometric study of Episodix (*Valladares-Rodriguez et al., 2017*) (i.e., content, face and ecological validity), the research discussed in this article was focused on criterion validity; that is, in the ability to identify cognitive impairment with statistical validity. For this, a collection of six ML algorithms widely used in medical research were selected according to their classification performance (*Lehmann et al., 2007*; *Tripoliti et al., 2010*; *Maroco et al., 2011*). More specifically, the ones applied in the mentioned preliminary study were also considered in this new research (i.e., Support Vector Machines (SVM); Logistic regression (LR); and Random forest (RF)) and three new algorithms were also introduced (i.e., Extra Trees classifier (ET); Gradient Boosting Classifier (GB); AB: Ada Boost Classifier (AB)), all of them run according to the default configuration, which in turn materializes one of the future research lines identified in the preliminary study. Furthermore, this new study introduces an additional improvement related to the training and testing data by introducing a 55-fold stratified cross validation procedure thanks to the richer dataset available. This mechanism prevents overfitting and artificial classification metrics due to the use of the same data for training and testing of classifiers, and dramatically improves the classical 80/20 data split strategy for training and testing. Moreover, despite of applying mean imputation to handle missing data, which may contribute to underrepresent the variability in the data, the impact in actual data set is limited because mean this value was computed by cognitive group and not for the all subject. As a consequence, variability was respected by cognitive group. Also,

the amount of missing data was low as it was only caused by different difficulty levels achieved by participants.

Attending to the classification results obtained with the first dataset in this new research including Episodix only, better results are obtained for GB, with an average value of $F1 = 0.93$ and a concordance Cohen's kappa $= 0.90$, over a maximum of 1. Besides, the recall and precision values obtained are all of them above 0.94, except precision for the MCI group (i.e., 0.79). These results demonstrate that Episodix has the ability to predict mild cognitive impairment, according both to the values computed for the harmonic average of precision and sensitivity (i.e., F1), and to the concordance of results when considering a random classification (i.e., Cohen's kappa). Results are also promising for ET and RF, followed by AB and SVM in decreasing order of classification quality. Worst results were obtained for LR, particularly precision for participants with MCI, as summarized in Table 2.

When applying the above ML algorithms to the second dataset (i.e., Episodix, semantic and procedural memory games), it can be observed that all metrics slightly improve. In this case, best results are obtained with the ET algorithm with $F1 = 0.97$ and Cohen's kappa $= 0.94$. Precision and recall values are also high, above 0.9 for all cognitive groups. With respect to the other ML algorithms, the next one would be ET, followed by RF, AB, SVM, and finally LR with slightly better results when compared to the ones obtained with the first dataset.

Note that results are sorted according to the Cohen's kappa index because it provides a measure of the validity of the classification when compared to a random result. Moreover, according to the standard interpretation of this index (*Cohen, 1960*), classification is almost perfect (0.81–1.00) for the complete dataset for all algorithms, except AB, SVM and LR in the case of the Episodix dataset, which can be considered as substantial (0.61–0.80).

Furthermore, we computed precision-recall curves as a useful measure of success of prediction when the classes are imbalanced. We plotted them as a multi-category class including average area and disaggregated by cognitive impairment (i.e., HC, MCI and AD groups). In information retrieval, precision is a measure of result relevancy, while recall is a measure of how many truly relevant results are returned. In other words, a high area under the curve represents both high recall and high precision in the prediction of cognitive impairment (i.e., participants affected by MCI or AD), where high precision relates to a low false positive rate, and high recall relates to a low false negative rate. Thus, as can be observed in Fig. 3, high scores are for ML classifiers based on ensemble methods (e.g., RF, GB, and ET, where precision-recall area is higher to 0.90 for all types of participants—HC, MCI or AD groups—). Thus, according to these outcomes and the statistical representativeness of the sample in Galicia, all of them allow to predict or discriminate cognitive impairment with high accuracy.

Finally, regarding the most informative features for classification, both using chi-squared and ANOVA $F$-value statistics methods, results were similar, these being the number of failures during immediate recall phase, short-term recall and long-term recall; number of omissions during long-term recall with semantic clues; number of repetitions during yes/no recognition phase; and time duration of procedural task. Particularly, we computed again

ML metrics using the reduced dataset. In general terms, the precision-recall area is higher using all the features (i.e., 81 features for Episodix and 100 features for Episodix+S+P) in comparison with the 10 most informative, including subject's age and genre. However, in the case of the ANOVA *F*-value selection and GB classifier, general results are equal, showing a slight decline in the prediction of MCI group (i.e., it varies from 0.90 to 0.86), but a better prediction in the case of HC and AD participants. These findings, using an 80% smaller dataset, have a future potential for a commercial product, where most likely computational restrictions would be a key issue. In the present pilot study, runtime of all experiments was the order of milliseconds.

To conclude, limitations detected during the preliminary study (*Valladares-Rodriguez et al., 2017*) were conveniently addressed and solved. In particular, with respect to accessibility, senior adults clearly preferred a touch interface better than traditional keyboard and mouse, and this was included in the new experiment. Another relevant aspect was the limited scope of the sample. In the present study, we have completed a statistically representative pilot experiment. Some improvements on the presentation speed of stimuli and the initial description of the tasks were made for this study. All of them contributed to further facilitate the administration of the game by professionals with different background, or even to further facilitate self-administration as a screening tool. Further research is needed to improve the resolution of the game for the identification of specific cognitive impairment conditions, as well as to achieve a complete assessment of the psychometric properties of the digital game, particularly the psychometric reliability.

## CONCLUSIONS

This study demonstrates the psychometric validity of the Episodix cognitive game. This is a clear step ahead in the introduction of serious games and machine learning for cognitive assessment, and therefore for detecting MCI or Alzheimer's disease; that is, to predict cognitive impairment using data captured from serious games. After overcoming the limitations identified during a preliminary study, a new statistically-significant study was carried out. Best prediction results were obtained with the introduction of extremely randomized trees and boosting trees (i.e., ET and GB) as classification algorithms. However, more trials are needed to achieve normative data with clinical validity. Finally, as a future line of research, the authors are exploring the introduction of item response theory (IRT) in Episodix in order to address some the limitations of classical test theory (CTT) about psychometric properties.

## ACKNOWLEDGEMENTS

We will like to thank the residence of Santa Marta of Vigo, the Association of Relatives of Alzheimer's Patients of Morrazo (AFAMO) for their support, and the Association of Relatives of Alzheimer's Patients and other Dementias of Galicia (AFAGA) for facilitating the selection of participants among its members for the present pilot study.

# APPENDIX A. LIST OF 10-FEATURES MOST INFORMATIVE OF THE EPISODIX DIGITAL GAME

**Table A.1** List of 10-features most informative of the Episodix digital game, usign chi-squared and ANOVA *F*-test.

| EPISODIX | |
| --- | --- |
| Chi2 | 1. RCL-LP2_Omissions: |
| | 2. RCL-LP1_Omissions |
| | 3. RI-B1_Failures |
| | 4. Rec-LP_Repetitions |
| | 5. RCL-CP1_Omissions |
| | 6. RL-LP_Failures |
| | 7. RI-A1_Failures |
| | 8. RL-CP_Failures |
| | 9. RI-A2_Failures |
| ANOVA *F*-value | 1. RI-A3_Omissions |
| | 2. RL-LP_Guesses |
| | 3. RL-LP_Omissions |
| | 4. Rec-LP_Repetitions |
| | 5. RL-CP_Failures |
| | 6. RI-A3_Failures |
| | 7. RL-LP_Failures |
| | 8. RI-A1_Failures |
| | 9. RI-B1_Failures |
| | 10. RI-A2_Failures' |
| **EPISODIX + semantic memory and procedural memory games** | |
| Chi2 | 1. RCL-CP1_Omissions |
| | 2. RL-CP_Failures |
| | 3. RI-A3_Failures |
| | 4. Rec-LP_Repetitions |
| | 5. RI-A2_Failures |
| | 6. procedurixTimeDuration |
| | 7. RL-LP_Failures |
| | 8. 8. RI-A1_Failures |
| | 9. RI-B1_Failures |
| | 10. RCL-LP1_Omissions |
| ANOVA *F*-value | 1.RL-LP_Guesses |
| | 2.RL-LP_Omissions |
| | 3.RL-CP_Failures |
| | 4.RL-LP_Failures |
| | 5.RI-B1_Failures |
| | 6.RI-A1_Failures |
| | 7.procedurixTimeDuration |
| | 8.RI-A2_Failures |
| | 9.Rec-LP_Repetitions |
| | 10.RI-A3_Failures |

# APPENDIX B. METRICS ABOUT PSYCHOMETRIC STUDY

**Table B.1** Metrics about psychometric predicted validity using the most informative features of EPISODIX dataset, sorted by Cohen's kappa coefficient.

| EPISODIX DATA SET | Chi-square | | | | | | ANOVA-F | | | | | |
|---|---|---|---|---|---|---|---|---|---|---|---|---|
| | | | Precision-Recall area | | | | | | Precision-Recall area | | | |
| | F1 | k↓ | HC | MCI | AD | ALL | F1 | k↓ | HC | MCI | AD | ALL |
| GB: Gradient Boosting | 0,87 | 0,82 | 0,85 | 0,65 | 0,89 | 0,79 | 0,87 | 0,83 | 0,87 | 0,63 | 0,89 | 0,8 |
| ET: Extra Trees | 0,86 | 0,81 | 0,85 | 0,66 | 0,86 | 0,79 | 0,84 | 0,79 | 0,83 | 0,6 | 0,86 | 0,76 |
| RF: Random Forest | 0,85 | 0,80 | 0,84 | 0,61 | 0,87 | 0,78 | 0,83 | 0,76 | 0,81 | 0,6 | 0,83 | 0,75 |
| SVM: support Vector Machine | 0,68 | 0,59 | 0,64 | 0,32 | 0,8 | 0,59 | 0,71 | 0,64 | 0,67 | 0,36 | 0,86 | 0,63 |
| AB: Ada Boost | 0,69 | 0,58 | 0,63 | 0,4 | 0,77 | 0,6 | 0,67 | 0,54 | 0,7 | 0,32 | 0,73 | 0,58 |
| LR: Logistic Regression | 0,59 | 0,52 | 0,63 | 0,22 | 0,76 | 0,54 | 0,62 | 0,56 | 0,64 | 0,24 | 0,81 | 0,56 |

Notes.

Episodix dataset (191 triples). Features: 10 most informative features, age and genre, using chi-squared and ANOVA $F$-value methods. ML algorithms: ET, Extra Trees classifier; GB, Gradient Boosting Classifier; AB, Ada Boost Classifier; SVM, Support Vector Machines; LR, Logistic regression; and RF, Random forest. Metrics: F1 score; Cohen's Kappa (i.e., used as index to order best classification); average and precision-recall area distributed by cognitive group. Cross validation (cv-fold = 55).

**Table B.2** Metrics about psychometric predicted validity using the most informative features of EPISODIX+S+P dataset, sorted by Cohen's kappa coefficient.

| EPISODIX+S+P DATA SET | Chi-square | | | | | | ANOVA-F | | | | | |
|---|---|---|---|---|---|---|---|---|---|---|---|---|
| | | | Precision-Recall area | | | | | | Precision-Recall area | | | |
| | F1 | k↓ | HC | MCI | AD | ALL | F1 | k↓ | HC | MCI | AD | ALL |
| GB: Gradient Boosting | 0,92 | 0,89 | 0,85 | 0,81 | 0,95 | 0,87 | 0,96 | 0,95 | 0,94 | 0,86 | 0,98 | 0,93 |
| ET: Extra Trees | 0,88 | 0,83 | 0,76 | 0,75 | 0,92 | 0,81 | 0,93 | 0,90 | 0,89 | 0,79 | 0,95 | 0,88 |
| RF: Random Forest | 0,88 | 0,82 | 0,75 | 0,76 | 0,91 | 0,81 | 0,93 | 0,91 | 0,9 | 0,8 | 0,95 | 0,88 |
| SVM: support Vector Machine | 0,71 | 0,60 | 0,56 | 0,46 | 0,81 | 0,61 | 0,74 | 0,65 | 0,56 | 0,49 | 0,87 | 0,64 |
| AB: Ada Boost | 0,68 | 0,58 | 0,56 | 0,38 | 0,8 | 0,58 | 0,69 | 0,60 | 0,54 | 0,42 | 0,81 | 0,59 |
| LR: Logistic Regression | 0,61 | 0,54 | 0,5 | 0,29 | 0,84 | 0,55 | 0,59 | 0,50 | 0,58 | 0,21 | 0,83 | 0,54 |

Notes.

Episodix, semantic memory and procedural memory dataset (458 triples). Features: 10 most informative features, age and genre, using chi-squared and ANOVA $F$-value methods. ML algorithms: ET, Extra Trees classifier; GB, Gradient Boosting classifier; AB, Ada Boost classifier; SVM, Support Vector Machines; LR, Logistic regression; and RF, Random forest. Metrics: F1 score; Cohen's Kappa (i.e., used as index to order best classification); average and precision-recall area distributed by cognitive group. Cross validation (cv-fold = 55).

**Table B.3** Metrics about psychometric predicted validity using the all features of datasets, sorted by Cohen's kappa coefficient.

| | EPISODIX DATA SET | | | | | | EPISODIX+S+P DATA SET | | | | | |
| | | | Precision-Recall area | | | | | | Precision-Recall area | | | |
| | F1 | k↓ | HC | MCI | AD | ALL | F1 | k↓ | HC | MCI | AD | ALL |
|---|---|---|---|---|---|---|---|---|---|---|---|---|
| GB: Gradient Boosting | 0,93 | 0,90 | 0,93 | 0,84 | 0,9 | 0,89 | 0,96 | 0,94 | 0,92 | 0,9 | 0,97 | 0,93 |
| ET: Extra Trees | 0,92 | 0,90 | 0,92 | 0,78 | 0,93 | 0,88 | 0,97 | 0,97 | 0,96 | 0,92 | 0,99 | 0,95 |
| RF: Random Forest | 0,91 | 0,88 | 0,88 | 0,76 | 0,93 | 0,86 | 0,98 | 0,97 | 0,97 | 0,93 | 0,99 | 0,96 |
| AB: Ada Boost | 0,81 | 0,76 | 0,76 | 0,52 | 0,91 | 0,73 | 0,84 | 0,80 | 0,72 | 0,61 | 0,95 | 0,76 |
| SVM: Support Vector Machine | 0,80 | 0,75 | 0,77 | 0,5 | 0,89 | 0,72 | 0,91 | 0,90 | 0,86 | 0,74 | 0,98 | 0,86 |
| LR: Logistic Regression | 0,76 | 0,70 | 0,72 | 0,42 | 0,89 | 0,68 | 0,84 | 0,81 | 0,73 | 0,6 | 0,96 | 0,76 |

**Notes.**

Episodix dataset (191 triples) and Episodix, semantic memory and procedural memory dataset (458 triples). All Features (e.g., 81 features for EPISODIX and 100 features for EPISODIX+S+P). ML algorithms: ET, Extra Trees classifier; GB, Gradient Boosting classifier; AB, Ada Boost classifier; SVM, Support Vector Machines; LR, Logistic regression; and RF, Random forest. Metrics: F1 score; Cohen's Kappa (i.e., used as index to order best classification); average and precision-recall area distributed by cognitive group. Cross validation (cv-fold = 55).

# APPENDIX C. SCREENSHOTS OF SERIOUS GAMES

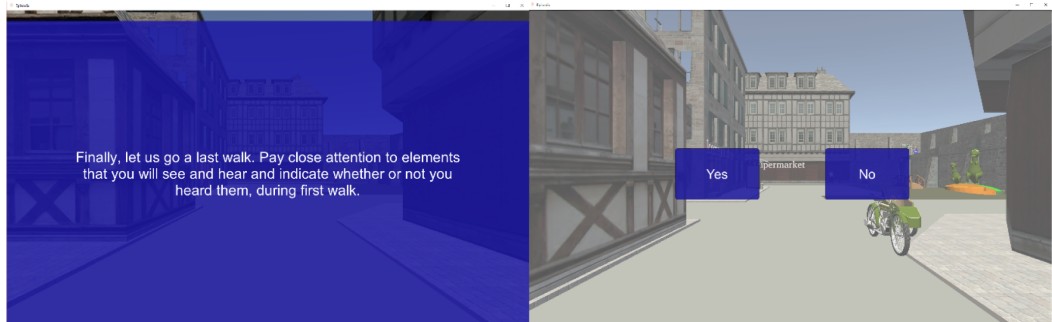

**Figure C.1** Screenshots of Episodix serious game, during yes/no recognition phase.

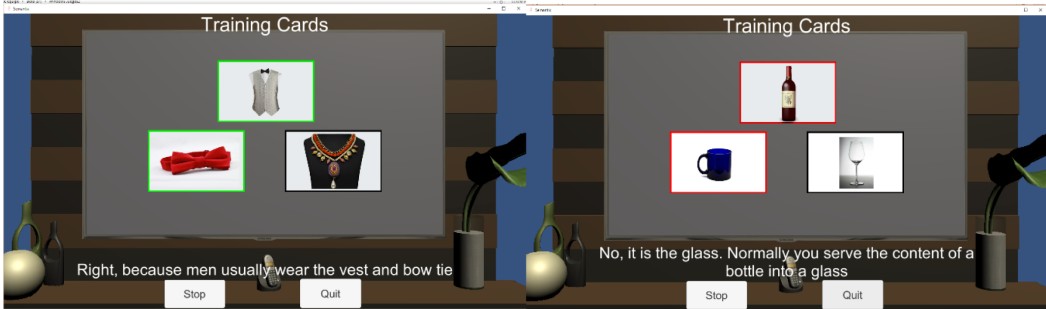

**Figure C.2** Screenshots of semantic memory game, during the initial training phase.

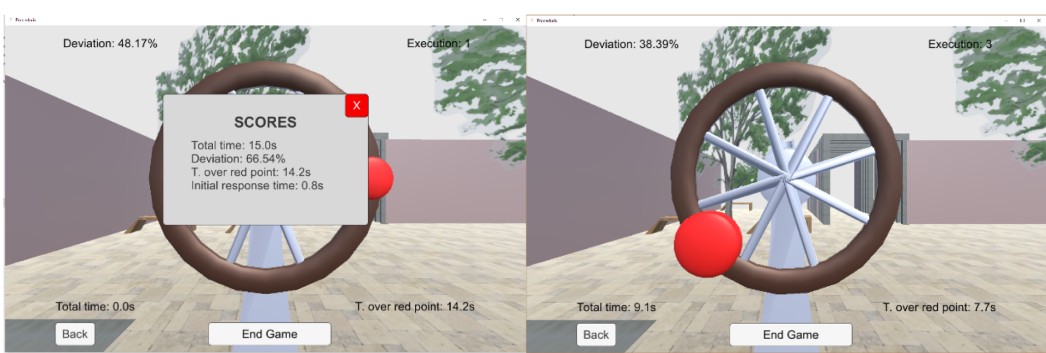

**Figure C.3** Screenshots of procedural memory game, after the 1 st execution and during the 3 th one, where ball's size varies.

### Funding

This research was supported by the University of Vigo. The funders had no role in study design, data collection and analysis, decision to publish, or preparation of the manuscript.

### Grant Disclosures

The following grant information was disclosed by the authors:
University of Vigo.

### Competing Interests

The authors declare there are no competing interests.

### Author Contributions

- Sonia Valladares-Rodriguez conceived and designed the experiments, performed the experiments, analyzed the data, contributed reagents/materials/analysis tools, prepared figures and/or tables, authored or reviewed drafts of the paper, approved the final draft.
- Manuel J. Fernández-Iglesias and David Facal conceived and designed the experiments, analyzed the data, contributed reagents/materials/analysis tools, authored or reviewed drafts of the paper, approved the final draft.
- Luis Anido-Rifón conceived and designed the experiments, contributed reagents/materials/analysis tools, approved the final draft, supporting funding.
- Roberto Pérez-Rodríguez approved the final draft.

### Human Ethics

The following information was supplied relating to ethical approvals (i.e., approving body and any reference numbers):

The study design was approved by the institutional ethics committee (i.e., Galician Ethics Committee for research (Spain), number code 2016/477) and was conducted in accordance with the provisions of the Declaration of Helsinki, as revised in Seoul 2008.

## Data Availability

The raw data are provided in the Supplemental Files.

## Supplemental Information

Supplemental information for this article can be found online at http://dx.doi.org/10.7717/peerj.5478#supplemental-information.

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
