# Peer review of "Episodix: a serious game to detect cognitive impairment in senior adults. A psychometric study"

_PeerJ, doi:10.7717/peerj.5478_

## Round 0.1 · original submission · Minor Revisions

Dear Author,

Numerous salient comments from the expert reviewers are enclosed to revise the manuscript. Please do these revisions so as the manuscript quality is improved

In particular the comments from Reviewers 1 and 4 must be addressed.

·

Basic reporting

The paper is clear in general, although some basic English errors were found:
- ...the next “none” (line 484) should be the next one.
- ...Genre (line 296) should be gender.
- TAM should be spelled out in full upon its first use (line 270)

In terms of literature references, herewith some issues and suggestions:
1. The reasons of increasing dementia phenomena should be mentioned, at least with a reference (line 35). Is it related to the advancement of detection technology?
2. “Organization” is not the last name of “World Health Organization” (line 39).
3. Relevant types of patient functional status (line 42) should be spelled out, e.g. physical, psychological, social and role.
4. The notion of “state-of-the-art” should be removed (line 18, 42, ), since the level of novelty in machine learning classification techniques (line 18) and diagnostic procedure (line 42) is relative and maybe subjective. In fact, the references used (Lezak, 2004; Howieson & Lezak, 2010) are more than five years old.
5. The notion of “novel” should be discarded (line 60) since novelty in video game will normally run out in a year.
6. The claim of being “fully” accurate is subject to criticism because the alpha set for statistical test was not absolute zero.
7. Limitations encountered during the preliminary study were not discussed between line 62 and 84. The discussion of the preliminary study was about methodological review, as opposed to specific limitations.
8. The notion of “recently” should be discarded (line 87) because the references used were over five years old.
9. Do “senior adults” (line 91, 96 & 453), “seniors” (line 168), “senior people” (line 180) share the same meaning as “older adults”? If so, it would avoid confusion to use either one of these phrases, else define each of them accordingly in Sect. 2.1.1 Sample description.
10. Is the Spanish version of CVLT (line 139) same as the Spanish version of CVT (line 137)?

As for the presentation of figs, the screens on tablet shown in C and D of Fig 2 are hardly visible for understanding the game features of semantic and procedural memory.

Experimental design

The psychometric criterion predictive validity should be the key focus of this study. However, the paper included results of a TAM questionnaire survey that collect participants' perceptions. These are indeed two non-complementary studies--one measured the predictive validity of Episodix, while another compared perceptions of three types of senior adults--with AD, with MCI, and HC. The former examined Episodix as a tool, while the latter used Episodix to examine perceptions of senior adults.

Only one research question (Is the Episodix game a valid tool─ with a statistical significance for older adults─ to discriminate between MCI, AD and healthy individuals?) was found in the paper (line 79-81), while the issues raised in the article were also dealing with usability of Episodix, which deserves a standalone research question.

In terms of participant recruitment, the usual onset of AD is 65 years old (Mendez, 2012), but the “older adults” involved in this study were over 55 years old (line 105-106), who were 10 years younger than the usual onset.

A definition of “older adults” as target users should be included with due justification in relation to the age of usual onset (line 110).

Technophobia was used as an exclusion criterion in the study without description on how the phobia was measured (line 111). Without the use of rigorous measuring tool (e.g. Hogen, 2009), excluding potential participants could be unethical.

A mixture of “subjects” (used 20 times) and “participants” (used 34 times) was found in this paper. This implies mixed methods research design but without any elaboration.

The count of “state-of-the-art” machine learning classification techniques (line 18) and the count of relevant metrics (line 19) should be stated.

.

Validity of the findings

The “best” classification result (line 22 ) should be limited to one as opposed to two (Extra Trees Classifier & Gradient boosting algorithm).

The types of limitations detected during the preliminary study should be mentioned in the Abstract (line 24).

The type of “additional research” (line 26) or “more research” (line 31 & 74) should be spelled out.

Additional comments

A study of psychometric criterion predictive validity of Episodix should not include a general usability study of Episodix. A comprehensive usability study of a serious games should cover at least the following five constructs: likeability, efficiency, control, learnability and helpfulness. Although TAM model is acceptable for examining the general use of serious games, rigorous usability instruments should be used to examine specific use of serious games.

Reviewer 2 ·

Basic reporting

Very interesting paper from the technological point of view and with therapeutic relevance

Experimental design

Appropriate methodology

Validity of the findings

Appropriate

Additional comments

Using games as an iinterview that mau also improve treatment for Alzheimer's disease

Reviewer 3 ·

Basic reporting

The paper reports on an empirical study around the validity of Episodix, a serious game, as a potential tool for discriminating among seniors who are health or suffer from Alzheimers or Dementia.

A major issue with the paper is that it never makes it quite clear what the difference is between the previous PeerJ paper and this one; as it is the game itself is not described in this paper (I assume it is described in the last paper) and this makes this paper a bit poorer.

Experimental design

1) Did each session with a senior happen in one single day? How long did the session last?

2) "Missing data was addressed by mean substitution by cognitive group (i.e., AD, MCI or HC)." What does this mean? What data was missing? Why?

3) What are the threats to validity?

Validity of the findings

The Results section repeats the numbers in the Tables - there is no need for that.

The Discussion section also starts by repeating some text from previous sections (on the sampling for example) and it does not go deep enough in the interpretation of the results and their implications.
How do the authors propose that the games be used?
What do the health professionals think?

Reviewer 4 ·

Basic reporting

The basic reporting components adhered to the PeerJ standard.

Experimental design

The main contribution of the article is on the psychometric and usability study of Episodix.

The psychometric study involved prediction using machine learning using Episodix labeled data. This was aimed to measure psychometric prediction validity. The author, however, did not expand on what does psychometric prediction validity really means.

Machine learning (ML) algorithms are predictive tools and they are black-box. They work based on the concept of supervised learning where data inputs were already labeled into classes to be predicted e.g MCI, HC, and AD. The algorithms then classify and return the performance metrics (termed as classification validity by author) such as accuracy, recall etc.

The concern was on the relatedness of psychometric prediction validity and classification validity. The author should explain how are they related or do they actually mean the same.

From the results reporting, it seems like the author used binary classification which distinctly separates between MCI, HC, and AD. Can overlapped cognitive condition happen? If yes, have author thought about using multi-class classification to do psychometric prediction validity?

The author later performed Feature Selection using chi-square and ANOVA-F which was a good step. Rather than simply concluding on the 10 most informative features, the author should re-run ML with those features and measure the accuracy; and further discuss how by reducing the number of features affect psychometric prediction validity? It could either increase or decrease the accuracy.

On the usability studies, TAM is an acceptable model to be used in the context since users are mostly registered patients. However, there are other models that could be mentioned and maybe more suitable that the author could review a bit. This is just an optional suggestion.

Looking at the questionnaire questions, I could acknowledge only one question as ATU. The question "I liked this game very much" doesn't fit as ATU. Has the author referred to any sample questionnaire using TAM before coming up with these questions?

The TAM questions were limited. A few more questions on ATU especially will give you a better indicator of the user willingness to use the game.

Validity of the findings

In the discussion section, the author mentioned that "participating senior adults felt comfortable during the evaluation sessions embedded in their daily routines", a strong conclusion without reference to any of the experiment instruments. I thought at first, this was concluded based on the TAM, specifically ATU's, questions but there were no questions on this.

The findings on ML are valid if they are based on binary classification.
However, it is still not clear what was the author intention on the psychometric prediction validity.

The author mentioned that EPISODIX is able to predict the MCI etc. But this really depends on the training set. ML in Episodix has to be trained first with sufficient data before prediction could be done. And as more users are added, the training set should be updated. Have the author thought about this? Yes, based on the EPISODIX data gathered and labeled, the prediction can be made quite accurately. How would EPISODIX work as a diagnostic tool when it relies on training data? Just some thought that came across my mind on this.

Overall, the findings are there but the storyline gets a bit distorted at the ML usage. There was no clear aim on what the psychometric prediction validity.

The conclusion has also not revisited the main contributions.

---

## Round 0.2 · accepted · Accept

Congratulations, Your revised manuscript has been accepted by PeerJ.

# ·

Basic reporting

All issues and suggestion highlighted in the previous review were addressed when revising the paper.

Experimental design

All issues related to the experimental design were addressed when revising the paper.

Validity of the findings

All issues related to the validity of the findings were addressed when revising the paper.

Additional comments

All issues and suggestion highlighted in the previous review were addressed when revising the paper.

Reviewer 4 ·

Basic reporting

Revised accordingly. The revised version is now recommended for publication.

Experimental design

Revised accordingly. The revised version is now recommended for publication.

Validity of the findings

Revised accordingly. The revised version is now recommended for publication.

Additional comments

Thank you for thorough explanation in the rebuttal.